# The Greek Urban Policy Reform through the Local Urban Plans (LUPs) and the Special Urban Plans (SUPs), Funded by Recovery and Resilience Facility (RRF)

Avgi Vassi [1,2], Konstantina Siountri [1,3], Kalliopi Papadaki [1,2], Alkistis Iliadi [1], Anna Ypsilanti [1] and Efthimios Bakogiannis [1,2,*]

1   General Secretariat of Spatial Planning and Urban Environment, Ministry of Environment and Energy, 115 23 Athens, Greece; a.vassi@prv.ypeka.gr (A.V.); ksiountri@aegean.gr (K.S.); k.papadaki@prv.ypeka.gr (K.P.); a.iliadi@prv.ypeka.gr (A.I.); a.ypsilanti@prv.ypeka.gr (A.Y.)
2   Department of Geography and Regional Planning, School of Rural and Surveying Engineering, National Technical University of Athens, 157 73 Athens, Greece
3   Cultural Technology and Communication Department, University of the Aegean, 811 00 Mytilene, Greece
*   Correspondence: ebako@mail.ntua.gr

**Abstract:** The lack of defined land uses in most parts of Greece (80%) has led to multiple environmental problems and phenomena of informal (arbitrary) construction with secondary side effects, such as a lack of basic technical and environmental infrastructure, unfair competition among private investors, the strengthening of climate change (increase in the number of urban diffusion) and the decline of natural and cultural resources. The Greek urban policy, over the last 100 years, has not succeeded in limiting these problems and for that reason the new Law 4759/2020 is expected to promote the development of a more efficient spatial planning system reform implemented through the Local Urban Plans (LUPs) and the Special Urban Plans (SUPs) that are funded by the Recovery and Resilience Facility (RRF). These programs will contribute to the preservation of cultural heritage and to the development of productive activities at both local and national levels, especially on the sectors of renewable energy sources, the circular economy, and the construction of "green" materials, digital applications and products etc. LUPs and SUPs are related to the holistic reform of the national urban policy and the relevant planning system that horizontally affects a wide range of policy areas such as: environmental protection and adaptation to climate change (for natural ecosystems and biodiversity; agriculture; forestry; fisheries; water resources; coastal zones), built environment and development, protection of historic sites and buildings, allocation of the public infrastructure, allocation of investments etc. The General Secretariat of Spatial Planning and Urban Environment Ministry of Environment and Energy has the main responsibility for the implementation procedures of all the proposed actions that will start in 2022 and will end in 2026. This paper focuses on the analysis of the current urban policy reform in Greece and the reasons that this reform is considered an immediate necessity in the current Greek urban legislative framework and the expected outcomes of LUPs and SUPs, which are examined in the literature for the first time, contributing to research on the present EU planning systems.

**Keywords:** urban policy reform; built environment; digitization; cultural heritage preservation; green transition; RRF

## 1. Introduction

Although European cities are in competition to gain investments for more than 25 years [1,2], Greek cities are among the least competitive in Europe. Indeed, they are not considered friendly to residents and visitors, as severe functional problems may be identified. Such examples include traffic congestion, environmental pollution, lack of environmental infrastructure, lack of green and open spaces, and degradation of the national cultural heritage.

At the same time, uncontrolled and continuous sprawl trends in the peri-urban and surrounding rural areas were identified more than half a century ago [3–6]. This phenomenon is quite obvious during the last years, not only in large metropolitan areas, such as Athens (Figure 1) and Thessaloniki—although measures have been set, such as the declaration of a Zone of Urban Development Control (ZUDC) in Attica, according to the Law 1337/1983 [7], but also in smaller towns and settlements [8]. It seems it is related to the lack of (a) defined land uses in most parts of the country (~80%) and (b) integrated urban planning measures and/or incentives. Regardless of the causes, its results are disastrous not only in terms of natural environment (habitats, landscapes, etc.) and resources (soil, water, forests, etc.) [9,10] but also in terms of cultural heritage (sites of historical value, traditional settlements) [11] and agricultural land and production (loss of valuable land in a time of food crisis) [12–15].

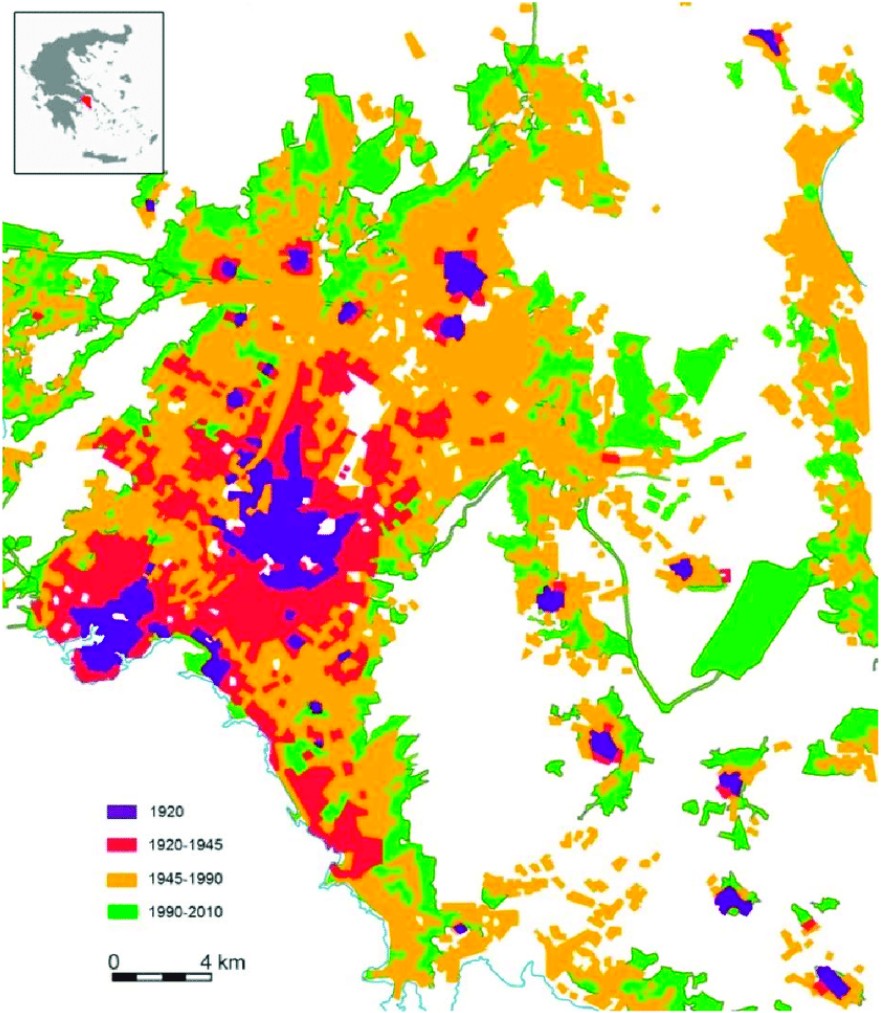

**Figure 1.** The urban development of Athens over the past 100 years [16].

Concerning cultural heritage, the lack of systematic and sustainable urban management is one of the major reasons leading to failures of protection and enhancement of the city's identity [17], as the aggressive development and policy deficiencies threaten cultural heritage properties [18,19], through the excessive privatization and commercialization of historic environment and public space due to lack of defined uses [20]. On the other hand, the management of cultural landscape plays a central role in the city planning, as in many cities the regeneration process starts from local tangible cultural heritage [21].

Over the last two years, the coronavirus pandemic affected all economic sectors in Greece, resulting in a general decrease of investment and consumption. Therefore, the need



for an extensive reform of the urban policy in Greece was considered an important step towards the quick recovery of the economy, the short-term mobilization of investments and the medium and long-term sustainable growth factor. The reforms consist of the Local Urban Plans (LUPs) and the Special Urban Plans (SUPs) and are based on the development of a streamlined system for solving the problems that have arisen in recent decades in the urban and spatial planning of Greece. The reforms are funded by the Recovery and Resilience Facility (RRF) and will be implemented in the period of 2022–2026 [22] by the Ministry of Environment and Energy.

The implementation of LUPs and SUPs was made possible by the new Law 4759/2020 "Modernization of Spatial and Urban Planning Legislation" [23] that led eventually to the simplification, acceleration, and improvement of the efficiency of the existing spatial planning system.

This paper provides a citation of the evolution of urban planning legislation in Greece that led to the necessity of the Law 4759/20 and an examination for the first time of the goals and the expected outcomes of LUPs and SUPs on the economy, the environment, and the heritage sites. The design of the procedures through which LUPs and SUPs will be implemented is also analyzed. The fact that their guidelines/plans to be implemented have not been thoroughly presented so far in the literature make this article eligible to contribute to the research of the EU planning systems [24]. Specifically in the field of cultural heritage, the dimension of urban planning in the protection of the cultural environment is also explored.

The paper is organized as follows: Section 2 refers to the methodology of the article, Section 3 provides (a) a retrospect of the Greek urban policy over the 20th and 21st centuries and the problems caused by the implementation of the Greek urban legislation of the past; and (b) a focus on the preservation of the cultural heritage through urban planning and design initiatives; Section 4 describes the new Greek planning legislative framework (Law 4759/2020); Sections 5 and 6 explain the details and the implementation of the new urban policy reform; Section 7 discusses the relative impact and Section 8 concludes our work.

## 2. The Methodology of the Research

The aim of the article is twofold: on the one hand, it focuses on recording the evolutionary course of the institutional framework of spatial planning in Greece, with the ultimate goals of: (a) investigating the planning options used by the legislator to solve key problems at the given time; and (b) the correlation of spatial and urban planning with the preservation and promotion of cultural heritage in urban centers. On the other hand, it is related to the thorough presentation of the latest institutional text, which aims at an integrated strategic planning at a local scale that will act as a coordinating framework for assembling a series of individual planning policies, such as the Sustainable Urban Mobility Plans and Electric Vehicle Charging Plans. The examined policies can be used as a tool for further research in the East Mediterranean area of EU or as a study case for future comparison of the current urban policy systems in Europe.

To achieve the above objectives, a combined methodological framework (Figure 2) was used where secondary information was studied and correlated. A central role in the methodology is played by the recording of the institutional framework and its evolution over time in the two fields of study (Figure 2, step 1): (a) spatial (emphasis on urban planning) planning; and (b) cultural heritage management. The two sectors are examined as two independent parameters due to the fact that the institutional framework for each followed a different course of development and with different starting points, while the basic tools that are utilized, as the case may be, show differences.

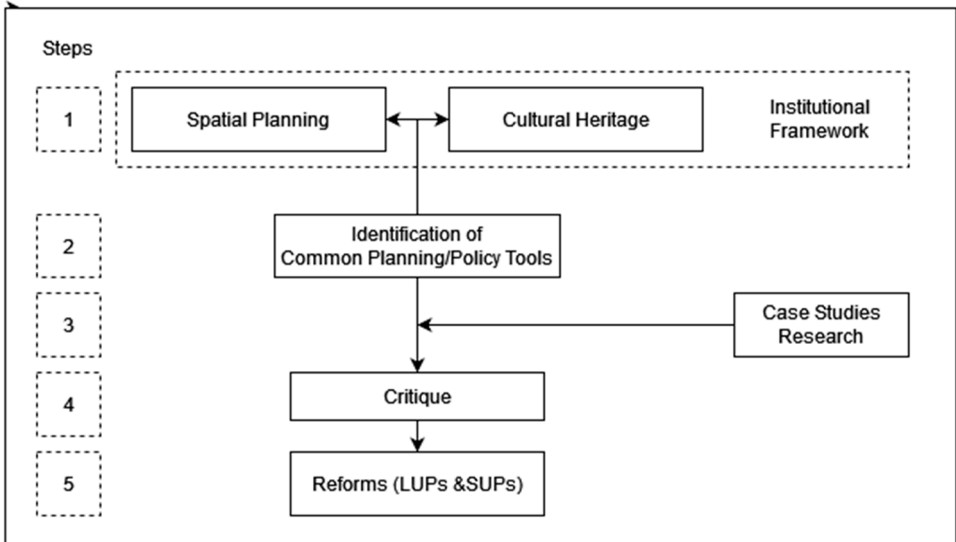

**Figure 2.** The methodological framework of the article. Source: Own Elaboration.

From the above separate actions, the common points of the two policies were identified (Figure 2, step 2), a fact that, in practice, translates into tools with a common point of reference both in urban planning and in the management and promotion of cultural heritage (Figure 2 step 3).

The following and final phase of the article (Figure 2, step 4 and 5) involved formulating a brief critique of the country's current institutional framework, focusing on the reforms of the Greek urban policy that are currently considered necessary for national spatial development.

The limitations of the research include the newness of the current institutional framework for spatial planning, which implies the non-implementation of a significant (if any) number of plans or studies, based on its directions. A consequence of this fact is the interest for future research in a deeper critique of the tools of the law and, in particular, in any problems or incompatibilities that may arise during its application. Finally, due to the large number of tools provided, the analysis and critique focus on the main tools, with the result that a future series of individual design policies is deemed necessary.

## 3. State of the Art

### 3.1. A Brief Literature Review

Spatial planning is linked to the institutional framework. By extension, the legal traditions of each country influence the way spatial policies are carried out as well as the result at the level of produced space. In fact, the spatial rules of law (especially, the urban, which are characterized by antiquity compared to spatial planning) have foundations in the legal systems of the respective countries or groups of countries. For example, a number of European countries, such as France, Germany, the Netherlands and Greece, belong to the thematic category of continental (civil) law. This particular category is based on codified rules based on primary sources and applied, by interpretation, when exercising control over urban planning policy.

Specializing, individually, in continental (civil) law in the European area, certain "legal families" can be found that are summarized in the following categories [25]: (a) German; (b) Scandinavian; (c) British; and (d) Napoleonic. Greece, although associated with the Napoleonic legal family, is an example of a hybrid institutional model since it draws its origins from both the Napoleonic and German schools: (a) it adopted France's centrally planned administrative system combined with polynomialism and bureaucracy that reduce the flexibility of the planning system; (b) the legal foundations of the German family were exploited, resulting in the identification of laws with a strict nature in their application.

As stated in the original theoretical hypothesis, the institutional framework interacts with planning traditions and systems, which implies an ambiguity regarding the influence of the two systems. The latter systems are classified by Newman and Thornley (1996) [25] into five groups called "administrative families:"

(a) German, based on the implementation of spatial policy at a number of levels, within distinct boundaries (legal, organizational, content). At the level of urban planning legislation, the role of the central state is special since it is exercised in execution of the dictates of the constitution. The role of the regions, which is strong, focuses on issuing the federal building code (BauGB) and the distribution of land uses (Baunutzungsverordung) [24];

(b) Napoleonic, which focuses on the establishment of a national code with urban planning regulations that operate at hierarchical levels of planning, of which the intermediate one (that of the district) is relatively weak with an emphasis on the national and local levels [26]. This fact is evident in the planning system of France, where the increased need to manage the hydrocephalus problem of Paris vis-à-vis the peripheral areas was managed in the context of the exercise of spatial policy on a national scale, utilizing the theory of development poles. At the level of regulations, pluralism prevails, a fact that is reflected in the 1995 law on spatial development and spatial planning, which did not limit the number of regulations but contributed to the addition of new ones with an interventionist nature [27];

(c) Nordic, which refers to a planning system with weak national and regional scales, which is associated with a strong local level of planning. Local government forms the detailed plans that affect urban development and licensing [26];

(d) British, which includes policies in which the role of central administration is supervisory to local administration responsible for the exercise of spatial planning. In terms of the nature of planning, in Britain the separation of physical planning from economic/strategic planning was applied, with the first type seeking to mitigate important phenomena such as the management of increased residential pressures, a subject to which the establishment of urban planning law rules is linked [28]. The preparation of land use plans in combination with the "development control" process were central tools that differed from the standard building rules [29];

(e) Family of eastern countries, characterized by countries with early systems of urban development. For this reason, this grouping seems vaguer with an emphasis on the polyphony found in the influences on the transforming systems.

Greece belongs to the Napoleonic planning system. Its system is characterized by the tendency to exploit legal rules of general application [25]. According to another version (EU Compendium 1997 [30]), the country's spatial planning system falls into another tradition (Urbanism Tradition) found in the Mediterranean countries of Europe and is characterized by strong architectural influence, interest in the urban landscape and design and strict building control rules. This strategy is confirmed through a brief review of the development models implemented in large European cities, such as Barcelona [31], Valencia and Genoa [32].

A common component, among others, was the identification of culture as an important factor for their development [33] and their emergence in the European urban system.

Coming back to the question of the two categories of families (legal and administrative ones), we can see agreement in the characterizations and geography, since countries with a specific direction in terms of legal rules tend to present corresponding choices in terms of the structure of their spatial planning system. Given the direct interconnection of culture with spatial planning, as noted above, through the examples mentioned at the local scale, an attempt is made in the following sections to record the two types of institutional framework in Greece (see Section 2).

*3.2. The Greek Framework of Spatial Institutional Planning*

The Greek urban policy, over the last 100 years focused (not successfully enough until recently) on the containment of the anarchic urban expansion and the introduction of regulations on the urban development and planning with the following legislative acts.

The first planning legislative framework was the Town Planning Decree of 1923 [34] that set the guidelines for the development and reformation of urban areas [35] by giving directions for implementing "Town Plans" (in the scale of a settlement of a town) [36]—one parameter in which emphasis was placed was land use determination [37]—and stressing the need for issuing building permits; as a result, it set the foundations for post-war urbanization in Greece [38]. As it was enacted quite shortly after the critical period 1922–1923—when 1.5 million Greek refugees from Asia Minor arrived in Greece, on a massive way [39] and, thus, urban development was unrestrained—it was seen as the general framework for urban planning policy across the country [40,41].

It was recognized as a quite progressive legislative document [42], as its directions were focused not only on housing management conditions (i.e., hygiene, aesthetics, safety, technological modernization) [43] but also on serving wider socio-economic goals (homogenization of a multicultural and multinational social capital [44] derived by the adaptation of the traditional "eastern" and pre-industrial architectural structures of the country's cities to the standards of Western European economic and residential development) [45]. According to this decree, a large number of Town Plans were formed for a wide range of cities and settlements of the country [46,47], which brought strong cultural elements (local architecture, traditional urban web, physiognomy of the place), which had been formed in recent centuries, mainly during the period of Ottoman rule, with these elements of the modern urban function of the 20th century.

However, these plans were often criticized [48] for their inability to organize the spatial, economic, and social development of the urban area in the long run. Moreover, as in all large European cities, the problem of urbanization and the viability of urban centers (pointed out by urban planners and architects in 1933 at the 4th CIAM (International Conference on New Architecture) the "Functional City" and ten years later published "Charter of Athens" [49], also affected Greece and especially Athens. The growth of the Capital led to the gradual abandonment of the rural areas.

This framework existed for more than 40 years [46]. During the 1970s, the spatial planning framework has dramatically changed [50]. The new era of spatial planning started in 1975 when the natural environment was recognized as a subject that was constitutionally protected [51,52] as a consequence of the state's international obligations [53,54].

This was validated by enacting the Article 24 (§ 1, 6) of the Constitution of Greece [55] according to which the natural environment and the cultural one should be protected as they are the rights of all citizen; in addition, a special regime for the protection of forests was established in Article 117 (§ 3 and 4) [55]. In Article 24 (§ 2–5) [55], provisions were included for the protection of the residential environment in order to serve the functionality of the cities and the residential areas in general and to ensure the best possible living conditions of the inhabitants. The provision of Article 24 [55] is considered innovative for its time. Concerning the protection of cultural heritage, and in accordance with the Constitution [55], the Ministry of Environment throughout the 20th century had gradually formulated a series of planning legislative documents which characterized more than 800 settlements as traditional and had designated as listed a large number of buildings.

Until 1979, spatial planning system in Greece consisted of a one-level study (town plans). This one-scale perception was transformed by enacting the Law 947/1979 according to which two hierarchical successive plans were necessary in order for urban space to be successfully planned. Apart from the settlement space, "land use plans" were also introduced at the sub-regional space [50]. This legislative framework has also established the pathway for active planning [38].

In 1983, Law 1337/83 [7] was institutionalized as a transitional law in order to fill the gap in planning legislation [56]. However, its implementation was expanded until the

late 1990s, as it was quite functionable. According to its provisions, two hierarchical levels of urban planning were proposed [7]: (a) the first one was about strategic planning in urban space; "General Development Plan" (GDP) was the planning tool which was applied in this level; (b) the second one focused on local level and it was implemented in two stages: the first was conducted through the "Urban Planning Study" (UPS); the second one had to do with the application of all the previous provisions and it included studies called "Urban Plans" (UPs). Law 1337/1983 has also activated another planning tool (Master Plan or Regulatory Plan) that came to the fore in 1972 when the Legislative Decree 1262/197 was institutionalized. In 1985, two Master Plans for Athens and Thessaloniki were enacted through ad-hoc legislative acts (Law 1515/1984 and Law 1561/1985).

In 1997, Law 1337/1983 [7] was updated by Act 2508/97 [57], as the new legislative framework functioned as a continuation of the urban planning regime introduced by the previous one [58]. According to the new perception, two "spatial levels" and four "planning steps" were proposed to be applied. More specifically, at the first level, the Master Plan was set as a first step in the planning procedure, while GDP was set a second one. At the same step, a new planning tool, called Plan for Spatial and Urban Organization of Open Towns (known as SCHOOAP), is incorporated in the planning system. This tool is about municipalities whose settlement populations did not exceed the number of 2000 residents. GDPs and SCHOOAPs were the basic planning tools of this period [59].

Subsequently, in 1999, Law 2742/99 introduced strategic spatial planning processes [46] in harmonization with the European Spatial Development Perspective (ESDP) framework [60] by providing two spatial planning tools at a national scale. These were the "General Framework for Spatial Planning and Sustainable Development", which was practically a national territorial plan and the "Special Frameworks for Spatial Planning and Sustainable Development", which were sectoral territorial plans for the whole country. Both levels were considered to belong mainly to the responsibility of the Ministry of Environment, Spatial Planning and Public Works (current Ministry of Environment and Energy) in terms of preparation, monitoring, evaluation and review [61].

Although the above-mentioned efforts to update and modernize the legislative framework for spatial planning, at both micro and macro scales, should be evaluated positively, it should be noted that previous acts had not been completely repealed. This policy has led to the development of a legal framework with provisions that are fragmentary and complex and required codification (EU, 1995) [62].

Recently, Law 4447/2016 [63] tried to reform the existing legislative framework [64] for developing a more flexible and responsive planning system [65], by promoting the application of previously introduced (Law 4269/2014) planning tools [40], such as Local Spatial Plans (LSPs) that have replaced the previous GDPs; another tool, called Special Spatial Plan (SSP), was also established. The LSPs are prepared in the whole territory of the first-degree local authorities while also inter-municipal LSPs may be prepared [63].

It is obvious from the above that Greek spatial policy places great emphasis on regulatory planning and spatial planning (Figure 3). However, there is a significant gap between the officially approved spatial plans and the reality, especially at the local level (Figure 4), as alternative informal mechanisms and greater flexibility in conforming to the law leads to a disparity between the formal laws and regulations and implementation [25].

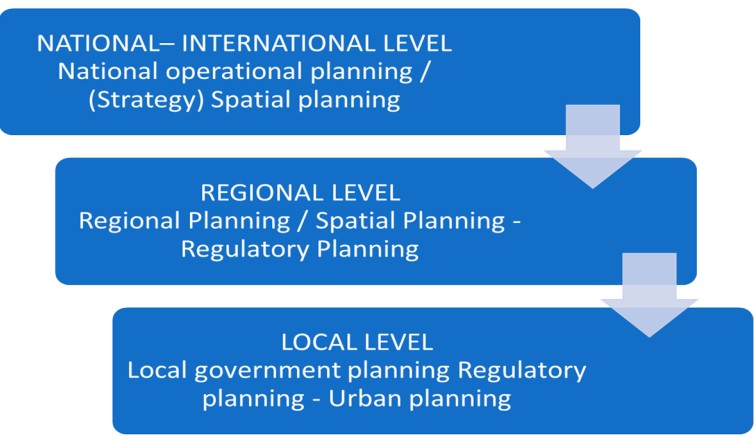

**Figure 3.** The levels of the Greek spatial policy.

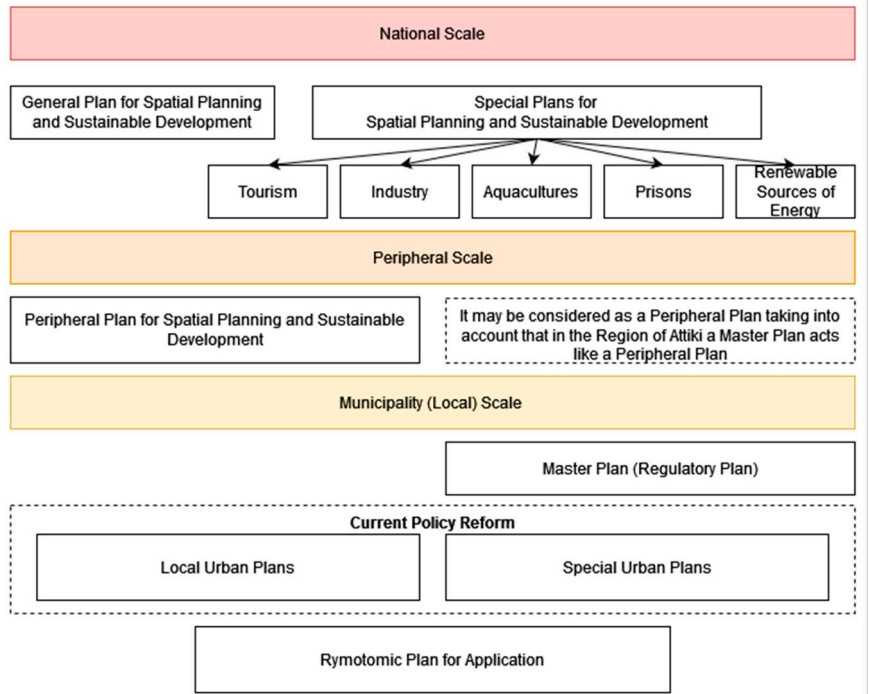

**Figure 4.** Diagrammatic presentation of the national approved spatial plans and the local (Peripheral and Municipality) level plans.

In the implementation of each reform, the problems that are recorded are the following [66]:

— The inability of the central administration to systematically guide the local authorities as well as planners and practitioners through the specialization of the principles and directions contained in the above planning and the institutional framework, as well as to monitor and systematically control the implementation of the desired policy (shaping standards and implementation guidelines and the development and ongoing updating of a monitoring mechanism);

— The variety of plans, tools and structures in spatial planning, their non-activation or their individual and fragmentary use;

— The inability to monitor the implementation of projects and the inability to identify and assess the impact of design options;

— The inability to respond and adapt urban plans to the ever-changing needs and data without the need to resort to ad hoc regulations and traditional and tried and tested practices;

－　The non-appropriate limitation of the variety of design levels, the unbalanced and clear wording of the binding of each level, the lack of simplification of the content of the studies and the approval procedures, as well as their technical and financial support;

－　The delays of investments and the further delay in meeting the required economic objectives of the state

*3.3. The Institutional Framework for the Preservation of Cultural Heritage*

According to the United Nations Economic Commission for Europe [67], spatial planning is an extremely important means of ensuring: (a) socio-economic development, adequacy and quality of environmental resources; and (b) protecting the natural and cultural heritage of an area, as well as and its limited natural and territorial resources.

The above reference to cultural heritage Is evident in cities with a long history where remnants of building complexes and traces of old road axes continue to exist and define their urban space. This specific finding refers to the case of Greek cities, where: (a) large-scale monuments of historical and archaeological value (of international, national and local importance) from various chronological periods are located, which are in direct interaction with urban life; (b) public buildings and residential buildings, of relatively recent periods, continue to function with their previous use or a different one; and (c) remains of industrial buildings and facilities are out of operation but refer to the historical past of the cities.

In Greece, despite the multitude of the above characteristic elements of cultural heritage reflected in the building stock of Greek cities (typical examples of such cities: Athens, Thessaloniki, Patras, Volos), the interest of urban planning in the preservation and highlighting of recent constructions (points b and c) was limited until the 1980s. Of catalytic importance was the enactment of the Presidential Decree of 1988 which provided for the preservation, repair or reconstruction of architectural, artistic and structural elements of listed buildings [68] and also the preparation and implementation studies of listed buildings of their ownership.

Already in the context of Law 2508/97 [57], provisions are found in the GDP for the planning of areas with a cultural imprint. However, a more important element is the reference to regeneration tools, i.e., interventions that solve the large building concentration and lack of public space, the coexistence of incompatible uses, the indifference to the cultural elements and activities of the areas, the existence of a built and natural environment in decline, the existence empty houses.

Other laws that followed, including Law 3028/2002 "On the Protection of Antiquities and Cultural Heritage In General" [69] and the New Urban Regulation (NUR) (Law 4067/2012) [70], although they addressed the issue, did not provide substantial solutions for better integration of the elements of the cultural landscape. While the Granada Convention was ratified by Law 2039/1992 [71], in the legislation that followed (Law 3028/2002 and Law 4067/2012) the obligations of the state are not defined. More specifically, no substantial incentives. i.e., grants, rehabilitation programs, tax exemptions, compensations, were given as the Presidential Decree provided for in article 48 of Law 3028/2002 has not been issued. Moreover, the blocking of the Building Right Transfer (BRT) tool has led to the desertification of many protected areas, especially those that need big scale renovation. It is pointed out that for listed buildings for which a BRT title is issued, the State "owes" the relevant "compensation" of transferring the rights to a zone outside the protected area. As a result of this, more and more holders of BRT to appeal to the Courts and lastly earn large sums of money.

These problems are particularly burdensome for the quality and attractiveness of the urban space, the perpetuation of the informal (arbitrary) construction, the devaluation of the existing building potential and the loss of the remarkable buildings and ensembles of the country.

As far as it concerns the landscape, the European Convention (Florence, 2000) [72] was ratified by the Greek Parliament only in February 2010, exactly ten years after its signing, thus covering a large gap, institutional and cognitive, in the field of Greek landscape

protection, constituting a particularly important text-tool for the protection of natural and structured environment. For most people the landscape is still some-thing vague, and its protection is still unknown and non-existent. The public and private acts of destruction of the qualities that characterize a place prove it, i.e., the uncontrolled construction, even in protected areas, the temporary constructions, the road opening, quarrying activities, the pollution, the destruction of wetlands, the fires etc. The main reason, of course, is the lack or the non-implementation of the spatial and urban planning.

The Greek cultural policy perpetually adapts and follows the European and international initiatives, uniform rules and laws. Concerning the protection of cultural heritage through land management, rural and urban development, since 1933 the Charter of Athens [49], identified that among the basic functional needs of a city design, a systematic planning direction of historic cities and settlements must be established. Later, the extensive catastrophes of the Second World War led the major cities of Europe (i.e., Warsaw, London, Leningrad, etc.) to take into deep consideration their historic background to their post-war regeneration.

In 1972, the "Convention for the Protection of the World Natural and Cultural Heritage" was signed in Paris" [73]. Both the contract and the corresponding recommendation concern all cultural goods, not exclusively buildings, but also spaces, as they establish new perceptions of the object and content of protection. The Declaration of European Architectural Heritage (Amsterdam Declaration 1975) [74] recognized the protection as an essential element of urban and spatial planning, through which, instead of the indiscriminate post-war reconstruction, the improvement of the urban environment should be promoted. The Granada Convention in Articles 10–11 in 1985 [71] proposes the inclusion of the protection of the architectural heritage as a direct objective purpose in the urban and spatial planning, acknowledging that not only the architectural heritage should be preserved for future generations, but that the appropriate functions and uses should be attributed to it.

In many cases the gradual creation and development of the European area was designed with a balanced element of spatial development and key objectives of the economic, social and spatial cohesion of the European area and influenced the respective national policies. In this context, spatial planning has been a catalyst in relation to the preservation and management of the architectural heritage, leading the European governments to formulate specific strategies for managing and controlling urban development [75].

Many Western European metropolises and large cities of the post-industrial era in the 1980s, invested in large projects that aimed at maximizing their attractiveness, e.g., urban regeneration, in some cases, has been applied to declining zones in inner cities or to coastal sites (i.e., Paris La Defense business center, London Docklands etc.). As far as it concerns the protection of the rural areas and landscapes, in Great Britain, attempts were made to define "green rings" around the expanding cities, while in the Netherlands, the protection of the large rural area enclosed between its four major cities (Amsterdam, the Hague, Rotterdam and Utter) was legally established.

Hafencity is a project of urban regeneration of burning interest, as both strategic and spatial planning that uses pilot methods of managing challenges, such as similar urban redevelopment projects in Europe, with the aim of enhancing Hamburg's attractiveness [76,77]. At the level of urban planning, the Hafencity Project is based on the following principles: (i) the new urban area to be attached to the historic center [77] and the absorption of the expansionist tendencies of the city; (ii) to have unique urban landscapes and therefore pluralism in the choice of types of urban composition design; and (iii) the city to develop in a symbiotic relationship with the river Elbe.

Another successful example of urban planning that also leads to the protection of cultural heritage is that of the Netherlands and, more specifically, of Rotterdam. Today, in the Netherlands, plans and policy texts allow planners at the two administrative levels to plan in open dialogue: (a) at the national level; (b) at the local level by municipalities (building plan, local land use plan, specific local plans e.g., for areas of regeneration or

improvement of living conditions). In Rotterdam, during World War II, 14.8% of the city's area was destroyed and 15% of homes were severely damaged. However, the bombing of the city was an opportunity to implement their comprehensive urban plan and dismantle many old buildings. In 1990, a major renovation of the port took place in which many new buildings were erected, and many old ones were renovated such as the Holland-America Lijn building. On the basis of this course was created the idea of the "creative city", which was the new trend of urban planning after 2000 [78]. Rotterdam eventually has become an example of European urban and architectural design with its continuous urban and architectural practice of experimentation and urban improvement.

## 4. The Greek Law 4759/2020 "Modernization of Spatial and Urban Planning Legislation" and the Innovative Reforms in Current Urban and Spatial Planning

The new Law 4759/2020 "Modernization of Spatial and Urban Planning Legislation" [23] is attempting the major reform of completing the spatial planning of Greece, which in the last 45 years has managed to reach only the 20% of the national territory. It also attempts the simplification, acceleration, and improvement of the efficiency of the existing spatial planning system.

The recent law improves the regulations for LUPs and SUPs and at the same time clarifies the relation of local urban tools with special planning tools. Particularly important is the introduction of the possibility to review a local urban plan ten years after the assignment of the study for its elaboration, as presumably this study will have become, after a decade, out of date. The Ministry of Environment and Energy has made an unprecedented and gigantic effort and has started the process for the preparation of LUPs and SUPs throughout the country.

The Law 4759/2020 [23] allows the development of a more efficient spatial planning system that affects horizontally a wide range of policy areas such as environmental protection and adaptation to climate change (for natural ecosystems and biodiversity; agriculture; forestry; fisheries; water resources; coastal zones; traditional settlements etc.), built environment and development or allocation of the public infrastructure and private investments, etc.

The elaboration of the above-mentioned relevant urban planning studies and their institutionalization aims at resolving the phenomenon of the confluence of multiple urban planning legislation that may also contain contradictory or ambiguous provisions, with a holistic approach and their introduction, a single text, of all the urban planning rules that concern an area. Thus, legal certainty will be ensured to their implementer.

In the context of drafting LUPs or SUPs, several issues are re-examined if deemed necessary by urban planning and the building conditions can be modified or supplemented. As a far as it concerns the protection of cultural heritage, indicatively, we refer to Article 7, § 9 [23], according to which limits, and regulations of presidential decrees issued may be supplemented or amended by the presidential decrees. The above regulation refers to typological architectural rules in protected settlements and cities, as well as to the modification or revision of the current road plan, even if it reduces the surface of its common areas for the protection, restoration, preservation and promotion of the urban fabric of traditional settlements, historical sites and archaeological sites, which is a component of their special physiognomy. The new law also disengages the crucial tool for the preservation of the architectural heritage—the Building Right Transfer (BRT)—from the problems by the Council of State that had rendered it inactive.

## 5. The Urban Policy Reform in Greece in the Framework of RRF

The next generation EU initiative, and in particular the Recovery and Resilience Facility (RRF), focuses mainly on promoting a green transition, increased public investment in urban regeneration, urban energy upgrading, and reducing the urban environmental footprint. Greece may already be considered as a country that gains initial benefits derived by development; this results from its planning agenda whose main interest is given in large

projects and strategic investments that will transform natural and constructed landscape, as happened in similar cases abroad [79,80] or expected to happen when similar projects are conducted.

The urban policy reform will be implemented in the framework of RRF by the General Secretariat of Spatial Planning and Urban Environment of the Hellenic Ministry of Environment and Energy. It will contribute to the green transition and is related to the following climate and environmental objectives defined by the EU 2020/852 Regulation [81] and the Convention for the Protection of the Architectural Heritage of Europe (1985) [71].

The proposed reforms correspond to the "local action plans" of the national strategic guidelines since environment (built and natural) represents the immediate field of intervention of them [82]. These actions aim to support the transformation of Greek cities towards climate neutrality, contributing to the implementation of the UN Agenda 2030 and the Sustainable Development Goals (SDGs), as well as the European Green Agreement [83] by 2050, for the improvement of the citizens' life.

Beneficiaries of the proposed investments are both the public and private sector as well as the individual property owners and more specifically:

- The municipalities involved, since the proposed interventions will upgrade and protect their space and will have institutionalized land uses in their administrative boundaries/territory;
- The Sectoral ministries and private investors, as they will know in advance which activities are allowed to be development in every place (so that to prepare properly their projects;
- The construction sector in the implementation stage of the interventions;
- The individual property owners and the users of the urban space in general.

*Description of the Urban Policy Reform*

The urban policy reform is related to the simultaneous coverage of the Greek territory with urban plans, and this is the first time in Greece that plans of the same level with the same technical requirements will be implemented in such a large percentage of Greek territory (Figure 5).

This was made possible by the legislative adjustments that needed to be implemented so as to modernize the planning legislation and by the technical specifications of studies that were enacted in view of this program so that the studies: (1) meet the latest political priorities of the EU; but also (2) contribute to the technological modernization of the state (geospatial databases compatible with the Single Digital Map).

At this juncture, more than ever, the following were deemed necessary:

(a) Spatial planning to provide a coherent framework that will contribute to improving the country's development prospects in terms of sustainability and prioritizing the safeguarding of the public interest. This direction required the rationalization of the spatial planning system and the improvement of its implementation at all levels, in terms of the sustainable management of land, rural and urban development;

(b) The rapid implementation of urban planning, before it becomes obsolete and inefficient, in combination at the same time with the protection of the architectural heritage, the historical centers, the preserved complexes, etc.;

(c) The activation of mechanisms and financial tools that will strengthen the operational capacity of the local self-government to implement the urban planning and urban regeneration programs;

(d) Local authorities to acquire an essential role by participating in the audit-approval process and in addition the opportunity to participate effectively in rehabilitation interventions of their areas or other local authorities, to give directions, to expedite procedures;

(e) Urban planning must be actively involved in the protection of the architectural heritage and its preservation within the urban fabric, through the activation of special building conditions and the harmonization of the proposals of the urban plans with the principles of the Valletta Convention [84] (the LUPs as an institutional tool to be

combined with the protection of monumental structures and historical ensembles by the Archaeological Law N. 3028/02 [69]).

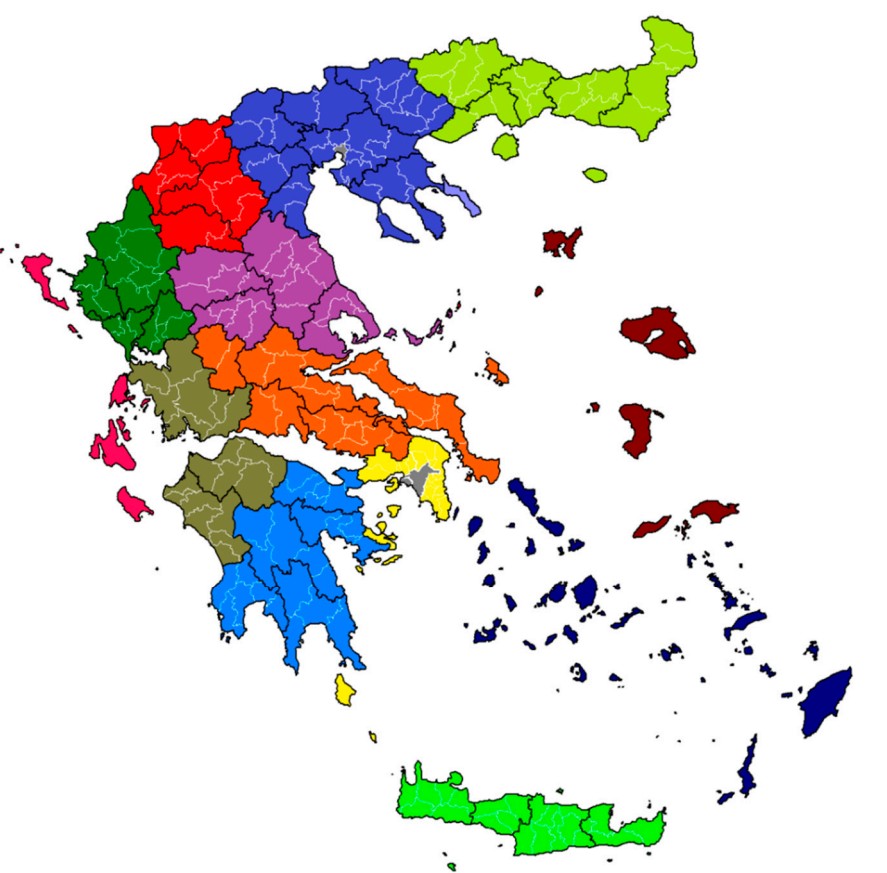

**Figure 5.** The administrative division of Greece. This image shows the 13 Regions, the 51 Prefectures and the 332 Municipalities of Greece (source: https://en.wikipedia.org/wiki/Administrative_divisions_of_Greece (accessed on 4 July 2022).

The Ministry is in charge to set the strategic and policy objectives that have to be achieved by these actions and also to design and oversee the implementation cycles. In this context, the Ministry and more specifically the General Secretariat of Spatial Planning and Urban Environment will determine the content and the technical specifications of the plans, will estimate the cost of each assignment cycle, will select the areas (municipalities, municipal units) which will be covered at each implementation phase and will provide the final approval of the content for each plan. After the completion of each plan the Ministry will promote the ratification of it by presidential decree.

The urban policy reform project, consist of five (5) actions (Figure 6), the two main actions of LUPs and SUPs that provide first grade planning and the three supplementary actions of Development Rights Transfer Zones (RTZ), of the Delimitation of Settlements and of the Plans for the characterization of Municipal Roads that provide second grade planning and consist a direct means of protecting the tangible cultural heritage and the physiognomy of the small settlements, the rural sites and the landscape.

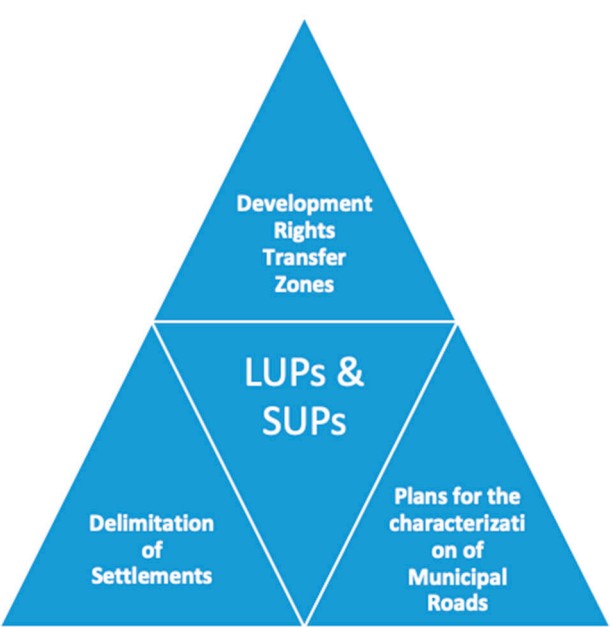

**Figure 6.** The Local Urban Plans (LUPs) and the Special Urban Plans (SUPs) consist of five (5) actions.

1.  Local Urban Plans (LUPs) (Law 4759/2020, Article 10) [23]

These plans will be prepared at the level of municipal unit for all municipalities in the country and through them will be institutionalized: land uses, building terms, regulations and restrictions, residential areas (existing settlements, plan extensions or new developments) which may also include private urbanization schemes, delimitation of settlements, definition of protection areas, areas for the development of productive activities (tourism, renewable energy resources, industry, agriculture etc.), important urban planning interventions, rural development, the implementation of Rights Transfer Zones (RTZ), areas of special urban incentives (e.g., to facilitate the allocation of large investments), road network, transport, construction and environmental networks and infrastructure, measures to adapt to climate change, measures to support emergencies and manage the consequences of natural and technological disasters and other threats, and any other measures, conditions or restrictions required for the integrated spatial development and organization of the area each LUP will cover (which is defined minimum at the level of municipal unit).

2.  Special Urban Plans (SUPs) (Law 4759/2020, Article 11) [23]

Their content is the same as the content of the LUPs but because it is a special urban planning tool the area in which an SUP is applied does not have to be identical (at least) to the administrative boundaries of the municipal unit applicable to LUPs. This practically makes the SUPs particularly useful and safe for the spatial organization and rural development of areas, regardless of administrative boundaries that can act as hosts for plans, projects and programs of supra-local or strategic importance, which require special regulation of land uses and other development conditions. SUPs can also be prepared for: (a) urban regeneration or environmental protection or disaster relief programs; and (b) critical spatial problems that require immediate treatment or prevention of completed situations due to lacking urban planning (in case of natural disasters like fires, earthquakes etc.).

3.  Development Rights Transfer Zones (RTZ) of Buildings (Law 4759/2020, Article 14 & 74) [23]

Preparation of this type of urban plans will define specific areas within the urban space (i.e., within approved urban plans or within areas where the urban planning process has started with a regulatory act or within approved settlement boundaries), which must be located outside the historic city centers and historic sites, traditional settlements, archeological areas, areas for which special building conditions have been imposed, etc. in order to increase the building efficiency of these areas. For the definition of these areas,

several parameters will be taken into account such as the degree of residential development, location, the physiognomy and the special characteristics of the area, etc. The definition of Development Rights Transfer Zones is an institutional precondition for the activation of another very useful cultural heritage preservation tool—the Building Right Transfer— which for decades has not been possible to implement. Defining these zones will make it possible to create public spaces in densely populated cities at low cost and at the same time to release the owners of thousands of land-plots or listed buildings from the constraints caused by the "special" characteristics of the properties and / or of the areas where these properties are located (e.g., historic centers).

4.     Delimitation of Settlements (Law 4759/2020, Article 12) [23]

This type of urban plans will determine the boundaries in settlements pre-existed of 1923 and have not been delimitated, as well as in newer settlements with less than 2,000 inhabitants that have not been delimitated but exist legally. The lack of settlements boundaries hinders or even makes it completely impossible for legal construction activity in the respective areas. Also, it must be noted that the settlements without boundaries, or with boundaries that had been approved illegally constitute a large percentage in the total number of settlements, and for this reason they have been canceled by the Supreme Court.

5.     Plans for the characterization of Municipal Roads (Law 4759/2020, Article 14) [23]

Municipal roads serve all types of connection that a municipality or a municipal unit needs such as the connections between settlements, important concentrations of activities and large facilities, important landmarks, etc. The Supreme Court during the last three years has overturned the legislation which determines (till now) the conditionalities for the construction of buildings outside the urban plan areas and mainly these related to the obligation of a plot to have access to an officially characterized road. As a result, too many plots outside the urban plan areas, even of significant size (e.g., tens of acres), have become inappropriate for building construction. This matter concerns both unstructured plots in which significant investments were planned, as well as the expansion of existing investments, mainly in the tourism sector. To deal with this matter, the Law 4759/2020 determines a specific and simple procedure for the designation of these roads (until now there was a legal gap or requirements that in practically very few cases could be met). The Ministry of Environment will activate the rapid implementation of this process, especially in areas with intense development pressures, through which local road networks will be identified and characterized over a period of 12–14 months.

## 6. Implementation of the National Urban Policy Reform

The preparation of LUPs and SUPs is a very demanding process that requires very good planning, systematic monitoring and coordination of many stakeholders from different sectors. As the General Secretariat will have the supervision of all stages of planning, implementation and delivery of the projects, within the narrow time limits set by the RRF (all the proposed actions will start in 2022 and will end in 2026), the management of all stages of preparation at administrative, legislative and executive level is of high importance. For the implementation of the actions the Ministry will set up a central mechanism which will involve the following public bodies: the Ministry of Environment and Energy (supervising authority/operator), the Technical Chamber of Greece and the Municipality to which the plan refers. This mechanism may be supported by other public bodies whose opinion is useful and from whom there will be information inflows during the preparation of the plans even by consultants of the private sector. The cooperation of the above-mentioned bodies and the monitoring of the projects will be done through the "electronic register of spatial studies" which is a repository of spatial studies but also a monitoring and communication platform between the involved parties of each study.

All actions of LUPSs and SUPs are scheduled to be implemented gradually as following:

- Action 1 (LUPs) is going to cover around 200 municipal units or municipalities For each LUP, the stages of Analysis and Proposal will define several thematic layers or files

with vector information that will be used for the compilation of the study. The thematic levels will be accompanied by a list of properties incorporating all the descriptive information deemed necessary for their cartographic representation, as well as any other necessary descriptive element. All deliverables should be available with open standards and consider the provisions of Law 3882/2010 [85] and all legislation on public data and e-Government;

- Action 2 (SUPs) is scheduled by the end of 2022 to include at least 10 assignments/contracts for plans in areas that face urgent issues such as: preparation of projects of supra-local or strategic importance, projects of urban regeneration or environmental protection or disaster relief, etc.;
- Action 3 (Development Rights Transfer Zones-RTZ) is going to be implemented in one phase of assignment/contract. The cycle will include around 80 municipal units especially those with densely populated cities or/and include historic centers where the definition of these zones will allow the activation of another very useful planning tool which is called building coefficient transfer. The municipal units to be studied will be defined through multicriteria analysis;
- Action 4 (Delimitation of Settlements) is going to be implemented in three phases of assignments/contracts. The first phase is about 20 municipal units whose settlements delimitation is urgent because of the cancellation decisions of the Supreme Court. The second and third ones are going to include 50 municipal units each;
- Action 5 (Characterization of Municipal Roads): is also going to be implemented in three phases of assignments/contracts. Every phase will include around 100 municipal units and the emphasis will be given in areas with intense development pressures where the local road networks is urgent be to be identified and characterized in order to cover the residential or productive needs of these areas.

## 7. Impact on Employment, Economy, Urban and Rural Development, Preservation of Cultural Heritage and EU Strategies

Urban planning policy attempts to make cities more inclusive, resilient, safe and sustainable. A central land management and good governance that includes all reliable information of rights, restrictions and responsibilities can support economic, social, and environmental sustainability [86]. Many forms of illegal development can also be significantly reduced [87], the spatial stabilization of the territory can be improved, and the natural cultural and historical resources can be protected [88].

For those reasons, these actions are a crucial parameter to the acceleration of investments and the sustainable growth and wellbeing at both local and national levels as its implementation will clarify and regulate the institutional framework (land uses, building terms, regulations and restrictions, protection areas, areas for the development of productive activities especially on the sectors of renewable energy sources and circular economy, areas of special urban incentives, etc.) for the allocation and construction of all types of projects and infrastructures, the protection of environment and cultural heritage and the definition of the measures to mitigate or/and adapt to climate change.

The main objectives are to restore the "weaknesses" of the existing development procedures, to remove unnecessary obstacles which up to now create delays in the implementation process of various projects, infrastructures, investments and to transform the spatial/urban planning from an investment barrier factor and an inefficient environmental management mechanism to a promotor of the productive investments and the protection of the environment.

Regarding the financial usefulness of the urban policy reform, it is noted that:

(a) the implementation of the relevant studies and projects will support the construction industry at all levels (designers, manufacturers—contractors, industry, etc.);

(b) the promotion and implementation of actions will stimulate "green" entrepreneurship and "green" innovation and technology;

(c)     the expected improvement of the urban environment will entail the growth of other business activities related to city branding;

(d)     the long-term benefits of the cultural heritage preservation (energy saving/upgrading of the urban environment.

These actions will effectively deal with the pathogenesis of the Greek urban centers and they will contribute to the green and digital transition of urban areas, in compliance with Europe's digital policy and initiatives (such as the new Digital Europe program), and increase their resilience, especially through the use of renewable energy sources in buildings and public spaces, the use of clean and sustainable transport, the growth of urban greenery, the strengthening and protection of urban biodiversity, the reuse of natural resources in urban areas and functions, and modernization towards a clean and circular economy, etc. Furthermore, the proposed plans will define the measures that should be taken to mitigate and/or prevent the effects of climate change. For this reason, each plan will comprise: (a) a specific chapter referring to the adaptation measures necessary at the local (municipal and sub-municipal) level, especially regarding the rise of the sea level, the Urban Heat Island (UHI) phenomenon, the promotion of the sustainable mobility, and the control of the urban sprawl; and (b) a specific chapter which will include a "local plan for the disaster management".

For the natural and cultural environment, the proposed plans (among others) will define the "Protection Areas" and the permitted land uses in these areas as well as the specific terms, regulations, restrictions and measures which are necessary to be taken according to the character and the protection status of each area (including the regulations provided by other environmental or archaeological laws, forestry areas and other protected areas). There will also be defined areas/zones which are appropriate for the development of productive activities especially in the renewable energy sectors, agriculture, industry, and tourism.

For all these reasons, the proposed reform has horizontal dimensions since it significantly affects all thematic pillars of RRF in terms of licensing and other prerequisite approvals (e.g., permitted land uses, strategic environmental assessment or environmental approval, building terms and building license, etc.), which are linked to the allocation and implementation of the proposed investments.

## 8. Conclusions

In the current policy reform, a clear vision is sought for each city that will guide the planning by combining both strategic and point-based interventions based on an overall plan, and a variety of short term and long-term actions. The success of such an endeavor requires the best synchronization and the best possible cooperation between all design and implementation bodies and of course the appropriate and timely funding that is provided by RRF.

The Hellenic Ministry of Environment and Energy through the proposed Local Urban Plans (LUPs), the Special Urban Plans (SUPs) is determined to provide a solution to several issues such as:

1.     The natural resource management, as well as the increase of natural space and biodiversity in cities;

2.     The regeneration of degraded areas, the promotion of mixed uses and the residential development. based on the bearing capacity of the environment;

3.     The coordination of urban governance, the production of statistics on living conditions in urban centers, public awareness, networking etc.

This reform is going to be a crucial parameter for the acceleration of economic and sustainable growth and wellbeing, at both local and national levels, as its implementation will contribute to the rapid economic recovery that requires public support for priority investments, especially in terms of improving the quality of the country's urban districts and settlements, as the urban environment is directly related to the Greek tourism product.

The proposed method of urban planning aims to alleviate spatial and social inequalities created by the degraded urban environment and the effects of climate change (e.g., thermal islands, energy poverty, deterioration of the building stock etc.) and to contribute to the development of productive activities at both local and national levels, especially in the sectors of the green economy.

Finally, these forms of integrated and long-term urban planning land use will contribute to the preservation of cultural heritage and the cities' cultural identity in a holistic way (old center, newer architecture, inland elements), as heritage is not limited to the historic centers and buildings, but also includes the newer constructions and the suburban areas.

It is estimated that, at the end of 2026, almost a century after the first urban planning legislation, Greece will present a comprehensive urban planning plan for the whole country, which will provide holistic solutions to all current issues around the organization and protection of urban and rural environments, according to the contemporary EU and international initiatives and regulations.

**Author Contributions:** Project administration A.V.; writing—original draft preparation, K.S. and K.P.; data curation A.I. and A.Y.; writing—review and editing, funding acquisition, E.B. All authors have read and agreed to the published version of the manuscript.

**Funding:** National Technological University of Athens (NTUA).

**Institutional Review Board Statement:** Not applicable.

**Informed Consent Statement:** Informed consent was obtained from all subjects involved in the study.

**Conflicts of Interest:** The authors declare no conflict of interest.

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
