# Peer review of "The Greek Urban Policy Reform through the Local Urban Plans (LUPs) and the Special Urban Plans (SUPs), Funded by Recovery and Resilience Facility (RRF)"

_land, doi:10.3390/land11081231_

Round 1
Reviewer 1 Report
The quality of the article was significantly increased. It is an article at the interface of a review article and a research article. It is a good resource for spatial planning research across Europe. The state of this issue in Greece deserves our attention, Greece is an a priori source of not only European culture and civilization.
There is still a lack of graphic examples to support the argumentation, e.g. urban sprawl in larger cities - Thessaloniki, Athens, Preservation of Cultural Heritage sites... and similar, the authors are focused more theoretically, they probably do not have an active design background. In some places, the article is oversaturated with information, but this can be accepted for a review text.
Author Response
A graphic example to support the argumentation, about urban sprawl in Athens was added.
Partial corrective changes were made throughout the article, with the aim of making the text more accurate and understandable.
Reviewer 2 Report
The next (third) version of the article is better than its previous versions. Enriching the article with a methodological chapter and illustrations certainly improved the perception of the article. However, after carefully reviewing the responses to the reviews and the changes made, I conclude that the authors still need to improve their article. In my opinion, it gives the impression of being incoherent. It is still heavily focused on the Greek urban policy reform proposal. This is clearly indicated even by the abstract itself, which lacks the construction proper to an abstract. It is just a description of the context and an outline of the aforementioned reform. In Chapter 2, the authors write that the analysis was conducted in an international and European context. Actually there are references, but as if by force. The article lacks order, coherence. Chapter 6 is the most important, and the others form a context for it, as it were, although not everywhere readable.
To Fig.1 I found no reference in the text. Likewise to Fig. 3. I also have the impression that the content of Fig. 1. is incomplete. Where is the reference to the proposed reform?
Chapter 3.2. the authors title the case of Greece. And yet Greece is also concerned with the other chapters, also there are references to Greece in Chapter 4 or 2. In my opinion, this makes the perception of the article more difficult, the reader may have the impression of chaos. In general, the structure of the paper is still a little chaotic and overextended. I propose to simplify and organize it. Enrich the content with a discussion and comparison with other countries (references to publications).
Besides, I point out that the authors did not respond to all the comments from the review. The responses are repetitive and do not always address the actual comments of the reviewers.
Despite the criticisms, I still believe that the article can be published but on condition of improvement.
Author Response
Comment:
It is still heavily focused on the Greek urban policy reform proposal.
Response:
It is true that the paper focuses on the presentation of the current urban policy reform and the examination of its goals for the first time. The outcome of this article may be used as a tool for further research in the East Mediterranean area of EU or as a study case for future comparison of the current urban policy systems in Europe.
Comment:
This is clearly indicated even by the abstract itself, which lacks the construction proper to an abstract. It is just a description of the context and an outline of the aforementioned reform.
Response:
The necessary changes have been made at the abstract
Comment:
In Chapter 2, the authors write that the analysis was conducted in an international and European context. Actually there are references, but as if by force.
Response:
It has been clarified that the article involved formulating a brief critique of the country's current institutional framework, which is supported by the partial investigation of case studies in other European countries
Comment:
The article lacks order, coherence. Chapter 6 is the most important, and the others form a context for it, as it were, although not everywhere readable.
Response:
Partial corrective changes were made throughout the whole article, with the aim of making the text more accurate and understandable.
Comment:
To Fig.1 I found no reference in the text. Likewise to Fig. 3. I also have the impression that the content of Fig. 1. is incomplete. Where is the reference to the proposed reform? Response:
Reference of Fig.1 existed. However, both of the pre mentioned figures have been improved
Comment:
Chapter 3.2. the authors title the case of Greece. And yet Greece is also concerned with the other chapters, also there are references to Greece in Chapter 4 or 2. In my opinion, this makes the perception of the article more difficult, the reader may have the impression of chaos. In general, the structure of the paper is still a little chaotic and overextended. I propose to simplify and organize it.
Response:
The structure and the content of the article was partially changed (and decreased) so as to make it more comprehensive
Comment:
Enrich the content with a discussion and comparison with other countries (references to publications)
Response:
As it is already mentioned paper focuses on the presentation of the current urban policy reform, which is supported by the partial investigation of case studies in other European countries. The comparison with other countries, in order to be effective, requires an extended analysis that can part of a future research work.
This manuscript is a resubmission of an earlier submission. The following is a list of the peer review reports and author responses from that submission.
Round 1
Reviewer 1 Report
This paper discusses several vital plans to be implemented by the future Greek government for Greek urban policy reform. Still, unfortunately, I don't think it can be published in a high-quality international journal like LAND.
First of all, the logic of the whole paper is not clear, and it is not organized in the usual academic paper format, which makes it impossible for the reader to understand what the author intends to do. In addition, the whole paper focuses exclusively on the Greek context, which I think does not meet the requirements of an international journal (in other words, it would be more appropriate to publish it in a Greek national journal). Finally, the authors do not clearly state this research's innovative and scholarly contribution, which is unacceptable.
In general, this paper is more like a government report than an academic paper, and therefore I reject it for publication.
Author Response
- The keyword Covid-19 was replaced by the keyword RRF as the programs of the LUPs, the SUPs and the Strategic Urban Interventions, funded by the Recovery and Resilience Facility (RFF).
- Changes on the structure of the text have been made
- The aim of the work, the scope of the research and the research questions have been added.
- Similar examples in Europe besides Rotterdam is given.
- The mention of the European Landscape Convention has been added
- References have been enriched
- Some repetition of content in the paper has been excluded.
- Unfortunately, there are not results (or study cases) of this procedure (presented for the first time in literature) that was initiated by Law 4759/2020 and funded by RRF. The first Municipalities started implementing their Plans the last few months according to the instructions that are described in the paper. The fist study cases will come up by the end of 2025.

Reviewer 2 Report
The article raises the general problem of spatial and urban planning. The relationship between the macro and local scales is a problem not only in Greece. The projection of political decisions into their professional implementation is reflected here.
The authors present a good overview of the history of the relevant regulations, somewhere in line 100. the mention of the Athens Charter of 1933, which influenced urban thinking for decades, would be appropriate. Although we know that it was not formulated in Athens, there is a certain symbolic connection.
Of course, the authors emphasize sustainability strategies in all three of its pillars, with an emphasis on cultural sustainability.
Line 250., why the example of the Netherlands and Rotterdam was chosen? Please explain. The example seems "lonely", out of context. There would be more similar examples in Europe.
The article would be helped by a case study that would demonstrate the application of the hierarchy of Local Urban Plans, Special Urban Plans, etc. It would be a projection of The levels of the Greek spatial policy on a specific situation (fig.2). The case study would support the research nature of the article.
Author Response
- The keyword Covid-19 was replaced by the keyword RRF as the programs of the LUPs, the SUPs and the Strategic Urban Interventions, funded by the Recovery and Resilience Facility (RFF).
- Changes on the structure of the text have been made
- The aim of the work, the scope of the research and the research questions have been added.
- Similar examples in Europe besides Rotterdam is given.
- The mention of the Athens Charter of 1933 and the European Landscape Convention has been added
- Citation is enriched
- There is not an available a case study that demonstrates the application of the hierarchy of Local Urban Plans, Special Urban Plans, etc., because they are now being prepared by the Municipalities.
- Images have been added.

Reviewer 3 Report
The presented article is interesting. It is a very specific work and can almost be interpreted as a case study. It is well structured, although it is not very appealing to the reader due to the lack of illustrations. The two illustrations included are somewhat "boring". Diagrams, images or graphic elements could be generated to allow a better understanding of the work for scholars and readers.
Author Response
- The mention of the Athens Charter of 1933 has been added
- Unfortunately, there are not results (or study cases) of this procedure (presented for the first time in literature) that was initiated by Law 4759/2020 and funded by RRF. The first Municipalities started implementing their Plans the last few months according to the instructions that are described in the paper. The fist study cases will come up by the end of 2025.
- Images have been added.

Reviewer 4 Report
The reviewed paper is a presentation of interesting solutions for urban policy reform in Greece. It has a certain value, but unfortunately it is not a typical scientific article. It does not have the structure of a scientific article, it is not clear what is the aim of the work, what methods were used, what was the scope of the research. What were the research questions formulated? Moreover, references to the literature are very rare, key international publications on spatial planning systems and urban policies (e.g. EU Compendium of Spatial Planning Systems and Policies; New Leipzig Charter; Newman, Thornley, 1996, Urban Planning in Europe...) are not cited, and there is no scientific discussion. It is good that the authors referred to the example of another European country besides Greece, namely the Netherlands. It is a pity that it is so brief. A comparison of the solutions applied in both countries would have been valuable. However, it is suggested to present a wider international context, e.g. France, Germany, Italy, England. Besides, I noticed some repetition of content in the paper, which should be excluded. Additionally, in the keywords and the introduction they mention the pandemic (COVID-19) but I did not find references in further parts of the paper. These should be added or omitted altogether. I can only guess that the proposed digitization may be related to it. However, it needs to be written openly, how does the proposed urban policy reform take into account the experience of the pandemic? It would also be interesting to relate the new solutions to the protection of the urban landscape (compare with the Unesco Recommendation on Historic Urban Landscapes or the European Landscape Convention).
The comments presented above do not support the recommendation of the article for publication. I would emphasize, however, that the paper has some value, but for it to be published it needs to be thoroughly improved. To sum up, I suggest: a thorough improvement of the structure of the paper so that it corresponds to the structure of a scientific article, an addition of references to the literature, a reference to a wider international context (perhaps a comparative analysis?). I would like to point out that not only Greece is currently reforming its spatial planning system or urban policy. Poland is among such countries. I also recommend reading an article on a similar topic: Gorzym-Wilkowski, W.A.; Trykacz, K. Public Interest in Spatial Planning Systems in Poland and Portugal. Land 2022, 11, 73. https://doi.org/10.3390/land11010073.
I am interested in the new version of the article, to improve it I strongly encourage.
Author Response
- The aim of the work, the scope of the research and the research questions have been added.
- The keyword Covid-19 was replaced by the keyword RRF as the programs of the LUPs, the SUPs and the Strategic Urban Interventions, funded by the Recovery and Resilience Facility (RFF).
- Changes on the structure of the text have been made
- The aim of the work, the scope of the research and the research questions have been added.
- Similar examples in Europe besides Rotterdam is given.
- The mention of the Athens Charter of 1933 and the European Landscape Convention has been added
- References have been enriched
- There is not an available a case study that demonstrates the application of the hierarchy of Local Urban Plans, Special Urban Plans, etc., because they are now being prepared by the Municipalities.
- Images have been added.

Round 2
Reviewer 1 Report
The author did not address my comments or even respond to them, so I chose to reject the paper.
Reviewer 4 Report
The reviewed paper is better prepared than its previous version. However, only some of the comments from the previous review have been taken into account. No significant changes have been made, as I suggested. Therefore, I have doubts whether I can recommend the article for publication in the international scientific journal Land. In particular, I emphasize that the paper does not have a typical structure of a scientific article, the purpose of the paper is still unclear, it is not described what methods were used, and I did not see a scientific discussion. What research questions were posed? However, I still believe that the content presented in the paper is valuable. In order for me to fully recommend the paper for publication, the authors should further refine the paper scientifically, according to the previous and current review. Please also consider the wider international context. Furthermore, the addition of Fig. 3 at the end of the article is puzzling. I suggest moving it to the beginning of the paper and adding a description of the administrative units. The title of Chapter 2. State of art is too general in relation to the content. It is unclear to replace Recovery and Resilience Facility with the abbreviation RRF and RFF. Which is the correct one?
I encourage you to improve your paper.